# Key Regulators of Angiogenesis and Inflammation Are Dysregulated in Patients with Varicose Veins

**DOI:** 10.3390/ijms25126785

**Published:** 2024-06-20

**Authors:** Daniel Zalewski, Paulina Chmiel, Przemysław Kołodziej, Marcin Kocki, Marcin Feldo, Janusz Kocki, Anna Bogucka-Kocka

**Affiliations:** 1Chair and Department of Biology and Genetics, Medical University of Lublin, 4a Chodźki St., 20-093 Lublin, Poland; pachmiel13@gmail.com (P.C.); anna.kocka@umlub.pl (A.B.-K.); 2Laboratory of Diagnostic Parasitology, Chair and Department of Biology and Genetics, Medical University of Lublin, 4a Chodźki St., 20-093 Lublin, Poland; przemyslaw.kolodziej@umlub.pl; 3Department of Neonatology and Neonatal Intensive Care, Independent Public Hospital No. 4 in Lublin, 8 Jaczewski St., 20-954 Lublin, Poland; marcinkocki@gmail.com; 4Chair and Department of Vascular Surgery and Angiology, Medical University of Lublin, 11 Staszica St., 20-081 Lublin, Poland; martinf@interia.pl; 5Department of Clinical Genetics, Chair of Medical Genetics, Medical University of Lublin, 11 Radziwiłłowska St., 20-080 Lublin, Poland; janusz.kocki@umlub.pl

**Keywords:** angiogenesis, inflammation, varicose veins, chronic venous disease, gene expression, plasma proteins

## Abstract

Varicose veins (VVs) are the most common manifestation of chronic venous disease (CVD) and appear as abnormally enlarged and tortuous superficial veins. VVs result from functional abnormalities in the venous circulation of the lower extremities, such as venous hypertension, venous valve incompetence, and venous reflux. Previous studies indicate that enhanced angiogenesis and inflammation contribute to the progression and onset of VVs; however, dysregulations in signaling pathways associated with these processes in VVs patients are poorly understood. Therefore, in our study, we aimed to identify key regulators of angiogenesis and inflammation that are dysregulated in patients with VVs. Expression levels of 18 genes were analyzed in peripheral blood mononuclear cells (PBMC) using real-time PCR, as well as plasma levels of 6 proteins were investigated using ELISA. Higher levels of *CCL5*, *PDGFA*, *VEGFC*, TGF-alpha, TGF-beta 1, and VEGF-A, as well as lower levels of *VEGFB* and VEGF-C, were found to be statistically significant in the VV group compared to the control subjects without VVs. None of the analyzed factors was associated with the venous localization of the varicosities. The presented study identified dysregulations in key angiogenesis- and inflammation-related factors in PBMC and plasma from VVs patients, providing new insight into molecular mechanisms that could contribute to the development of VVs and point out promising candidates for circulatory biomarkers of this disease.

## 1. Introduction

Cardiovascular diseases are the most common cause of death worldwide, with a growing trend in developing countries. In addition to high mortality, cardiovascular diseases cause a significant reduction in quality of life and disability due to complications [1,2,3,4,5]. Cardiac diseases such as coronary artery disease, ischemic heart disease, and myocardial infarction are the most intensively studied, while less attention is paid to peripheral vascular diseases. One of the most underestimated diseases associated with the peripheral vascular system is chronic venous disease (CVD). CVD is characterized by a high incidence rate of multifactorial etiopathogenesis and is burdened with serious complications, making this disease a major medical problem worldwide [6].

CVD is defined as a syndrome of chronic morphological and functional abnormalities of the venous circulation of the lower extremities caused by impaired blood flow resulting from incompetence of the venous valves, venous occlusion, or a combination of these factors. Hemodynamic disorders in CVD are exacerbated by muscle pump dysfunctions, especially with regard to the calf muscles [7,8,9,10,11,12,13,14,15]. CVD encompasses a wide spectrum of clinical presentations, such as telangiectasia, varicose veins (VV), leg edema, and skin changes. The most common manifestation of CVD are VVs, which appear as abnormally dilated and deformed superficial veins that become increasingly tortuous and enlarged. The prevalence of VVs among general practitioner attendees was estimated to be approximately 17% [16,17]. The symptoms that usually accompany VVs include tingling, aching, itching, pain, muscle cramps, and sensations of throbbing or heaviness in the legs. If not properly treated, the disease could progress to advanced stages, including skin changes and venous ulcers, leading to significant impairment of patient abilities and a reduction in quality of life [11,18]. Common risk factors for VVs include age, obesity, smoking, low physical activity, periods of prolonged standing or sitting, and a positive family history [16,18,19].

The recommended diagnostic methods for VVs are physical examinations of the lower extremities and imaging using color duplex ultrasonography or other techniques [9,19,20]. To establish a precise diagnosis, the disease is classified according to the CEAP (Clinical, Etiology, Anatomic, Pathophysiology) classification [9,17,18]. The most effective treatment options are compression therapy and invasive interventions, complemented by pharmacotherapy [19,21].

In VVs, hemodynamical disturbances in the flow of venous blood in the lower limbs lead to systemic or local venous hypertension, venous reflux, and microcirculation disorders. Venous hypertension causes progressive dilatation and weakness in the vein wall, leading to an abnormal pressure gradient in the vasculature. Consequent differences in shear stress transduce physical signals to the endothelium monolayer, increasing the expression of adhesion molecules and promoting leukocyte-endothelial activation. Endothelial dysfunction triggers an inflammatory response, exacerbated by leukocyte infiltration, cytokine production, and activation of matrix metalloproteinases, leading to weakening of the vein wall, fibrosis, and vascular remodeling, accompanied by venous valve dysfunction. Increased vein permeability and subsequent microcirculatory alterations create local hypoxic conditions that stimulate the growth of new blood vessels (angiogenesis) [7,8,9,10,11,12,13,14,15].

Increased inflammatory response and enhanced angiogenesis are considered important contributors to VVs progression, but there is little information on dysregulations in inflammation and angiogenesis-related signaling pathways in patients with this disease. The appearance of altered regulation of inflammation and angiogenesis in VVs suggests studies indicating altered levels of different MMPs in this disease [7,22,23]. Increased inflammatory response as a consequence of chronic damage to the vessel wall caused by turbulent blood flow was associated with an elevated level of IL-6 in blood samples taken from VVs [24]. Plasma levels of such proinflammatory factors as ICAM-1, VCAM-1, angiotensin-converting enzyme, and L-selectin were increased after blood stasis in VVs [25]. Furthermore, previous studies indicated dysregulations of genes associated with angiogenesis in VVs [26,27,28]. Higher expression of VEGF-A and VEGF-R2 was observed in varicose vein specimens [29]. Increased expression of other regulators of angiogenesis, including HIF-1α (hypoxia-inducible factor-1α) and metallothionein, was also demonstrated in varicose veins [30]. Decreased plasma levels of pro-angiogenic factors such as TNF-α and VEGF-C were observed in VVs patients after diosmin treatment [31]. Furthermore, certain SNPs in genes regulating angiogenesis (*VEGF*, *EFEMP1*, *CASZ1*) were associated with different stages of CVD [32,33,34].

A growing amount of evidence indicates that the abnormal course of inflammation and angiogenesis seems to play a crucial role in the venous pathology underlying VVs; however, the signaling pathways involved still constitute a large knowledge gap. Therefore, in the presented study, we analyzed patients with VVs and the expression of key factors that regulate inflammation and angiogenesis to better understand how these processes are driven in this disease. Furthermore, obtained data about the pathogenesis of this disease could also point to targets potentially useful in the prospective development of novel methods for the diagnosis and treatment of VVs.

## 2. Results

### 2.1. Characterization of the Study Subjects

Two groups of subjects were included: 40 patients with VVs (VV group) and 24 healthy volunteers (control group). The comparison of characteristics of the studied groups is presented in Table 1. No statistically significant differences were found between the groups in age, sex, body mass index (BMI), hypertension, hypercholesterolemia, and plasma levels of LDL, HDL, and C-reactive protein. However, VVs subjects had a higher mean level of cholesterol and lower mean levels of creatinine, urea, fibrinogen, and homocysteine in blood than the control group. Associations between these characteristics and the expression of the analyzed genes and proteins were explored and presented in Section 2.4.

### 2.2. Genes Related to Angiogenesis and Inflammation Are Dysregulated in VVs

The expression levels of 18 genes associated with angiogenesis and inflammation (*ANGPT1*, *ANGPT2*, *CCL2*, *CCL5*, *CSF2*, *CXCL8*, *FGF2*, *IL1A*, *IL1B*, *IL6*, *PDGFA*, *PDGFB*, *TGFA*, *TGFB1*, *TNF*, *VEGFA*, *VEGFB*, and *VEGFC*) were analyzed in peripheral blood mononuclear cells (PBMC) of 40 patients with VVs (VV group) and 24 healthy controls (control group) using the real-time PCR method. During the multistep data quality control procedure, four out of 18 analyzed genes (*ANGPT2*, *CSF2*, *IL1A*, and *IL6*) were excluded from the data due to receiving low signal or unreliable data in more than half of the subjects in at least one of the study groups (Appendix A). Furthermore, in two samples, low-quality data was found in more than half of the analyzed genes (Appendix A), as well as eight samples were identified as outliers (Appendix A); therefore, these samples were also excluded from the analysis (see the Section 4 for more details).

Differences in the expression levels of 14 retained genes between the VV and control groups were determined using the dCt method for relative quantification [35,36]. The distributions of normalized and 2^−dCt^ transformed expression levels of these genes in the studied samples are presented in Figure 1. Differences in the expression levels of the studied genes between the compared groups are represented as fold change values (Table 2). Thirteen of the analyzed genes (*ANGPT1*, *CCL2*, *CCL5*, *CXCL8*, *FGF2, IL1B*, *PDGFA*, *PDGFB, TGFA*, *TGFB1*, *TNF*, *VEGFA*, and *VEGFC*) showed higher expression in the VV group compared to the control group, while the expression of *VEGFB* was lower in this comparison. When tested for statistical significance, four out of the analyzed genes (*CCL5*, *PDGFA*, *VEGFB*, and *VEGFC*) were significantly differentially expressed (*p* < 0.05) between the compared groups (Table 2).

ROC analysis was used to evaluate the performance of classifying subjects into the appropriate group based on the expression of the analyzed genes. The highest values of areas under the ROC curves were obtained for four genes characterized by statistical significance (from 0.661 to 0.748), showing moderate classification accuracy (Table 2, Appendix A).

The differential character of the studied genes was further evaluated using logistic regression performed in both univariate and multivariate modes. The odds ratios (OR) with *p* values were calculated to illustrate the chance for the VVs condition to occur when the mean of 2^−dCt^ values in the studied samples doubled. Univariate logistic regression shows that only *VEGFB* had statistically significant OR values (Table 2 and Appendix A). In the multivariate mode of logistic regression, sex, age, BMI, and smoking variables were used to evaluate whether the differences in expression of the analyzed genes between the compared groups could be considered independent of these variables. Similarly to the univariate analysis, the differential expression of only *VEGFB* remained statistically significant after adjustment on these variables; therefore, this gene could be considered as independent (Appendix A). In turn, the expression of the remaining genes could be significantly associated with other variables; therefore, an analysis of such associations was performed, and results were provided in Section 2.4.

Obtained results indicate that increased expression of *CCL5*, *PDGFA*, and *VEGFC* and lowered expression of *VEGFB* in PBMC could be associated with VVs and probably illustrate dysregulations in angiogenesis and inflammatory pathways in this disease. This knowledge could be used to understand the pathology of VVs; however, the results presented need to be validated and deepened in further studies.

### 2.3. Angiogenesis-Related Proteins Are Dysregulated in VVs

The ELISA method was used to analyze plasma levels of six key regulators of angiogenesis (ANGPT-1, ANGPT-2, TGF-alpha, TGF-beta 1, VEGF-A, and VEGF-C) in 40 patients with VVs (VV group) and 20 healthy controls (control group). The amounts of samples in which the concentrations of the analyzed proteins were quantified or not are presented in Appendix A. The concentrations of TGF-alpha and TGF-beta 1 were below the detection range in the control group but were detected in almost all VVs patients. In turn, VEGF-C was below the quantification range in almost all samples of the VV group, but plasma concentrations of this protein were determined in all samples in the control group. Analyzed angiopoietins were detected in all analyzed samples. Hierarchical clustering and the PCA analysis showed the presence of two outlier samples in the data, which were excluded from the analysis (Appendix A). The plasma levels of six analyzed proteins in the studied groups are illustrated in Figure 2. The mean concentrations of TGF-alpha, TGF-beta 1, VEGF-A, and VEGF-C were statistically significantly different between the VV and control groups, and the ROC values obtained for these proteins ranged from 1.000 to 0.800. Regarding ANGPT-1 and ANGPT-2, the difference was not statistically significant, and the ROC showed low precision in the classification of the analyzed samples for the studied groups (Table 3, Appendix A).

### 2.4. Relationships with Risk Factors and Biochemical Parameters

Potential associations between the expression of four genes (*CCL5, PDGFA, VEGFB*, and *VEGFC*) or plasma levels of four proteins (TGF-alpha, TGF-beta 1, VEGF-A, and VEGF-C) obtained with statistical significance and clinical features (risk factors and biochemical parameters) were explored. Associations regarding continuous-type characteristics (age, BMI, and blood levels of cholesterol, LDL, HDL, homocysteine, urea, fibrinogen, creatinine, and C-reactive protein) were investigated using correlation analysis and univariate linear regression. No strong correlations were obtained for genes (|R| < 0.42), and those of statistical significance have negative correlation coefficients between urea blood levels and the expression of *CCL5*, *PDGFA*, and *VEGFC* (Table 4 and Appendix A). In the linear regression analysis, all these relationships were confirmed as statistically significant (Table 4). In relation to protein analysis, twelve statistically significant correlations were obtained (Appendix A), including six ones confirmed by simple linear regression (Table 4, Appendix A). Obtained results showed that urea blood levels could be associated with the expression of genes dysregulated in VVs; however, the correlations are rather weak. In turn, among the analyzed proteins, the highest number of associations were found for creatinine blood levels (associated with TGF-alpha, VEGF-A, and VEGF-C).

For categorical-type characteristics (sex, hypertension, and hypercholesterolemia), the evaluation of the relationships with the analyzed genes and proteins was performed using the Mann-Whitney U test and the Student’s *t*-test, depending on the normality of the data. Statistically significant differences were found in the expression of *VEGFC* (*p* = 0.0198) between patients with and without hypertension (Appendix A). Regarding proteins, plasma levels of TGF-alpha (*p* = 0.0358) were statistically significantly associated with hypercholesterolemia (Appendix A).

### 2.5. Coexpression of Selected Genes and Proteins

To investigate a similarity in the expression patterns in the studied genes and proteins, a pairwise correlation analysis was performed for four genes (*CCL5, PDGFA, VEGFB*, and *VEGFC*) and four proteins (TGF-alpha, TGF-beta 1, VEGF-A, and VEGF-C) selected as statistically significantly differentiating the VV group from the control group. The strongest positive correlations were found between expression levels of *PDGFA* and *VEGFC* (R = 0.75, *p* = 8.08 × 10^−10^) and between plasma levels of TGF-alpha and TGF-beta 1 (R = 0.81, *p* = 1.95 × 10^−12^) (Figure 3, Appendix A). It could indicate that the correlated factors are coexpressed functionally associated, and the altered activity of the regulated processes probably contributes to VVs development.

The interesting situation was observed in two other associations with a high correlation coefficient: between plasma levels of VEGF-C and proteins belonging to the TGF family (TGF-alpha and TGF-beta 1). In both cases, the correlation coefficients and the *p* values were the same (R = −0.80, *p* = 1.40 × 10^−11^). This is because, in the samples in which the VEGF-C concentration was determined, the levels of TGF proteins were too low to be quantified by used ELISA tests and were not determined, and vice versa. It suggests antagonistic relationships between plasma levels of VEGF-C and TGF with increased TGF levels in VVs patients; however, more studies are needed to validate this conclusion.

A weak and negative correlation was obtained between the expression of the *VEGFC* gene and plasma levels of encoded VEGF-C protein (R = −0.41, *p* = 3.95 × 10^−3^). It could be concluded that the expression of this gene in PBMC is not reflected in protein concentration in the plasma compartment.

### 2.6. The Expression of the Analyzed Genes and Proteins Are Unable to Predict the Localization of Varicosities

To assess whether the expression of 14 analyzed genes and 6 proteins could be indicative of the localization of varicosities, the differences in the expression of the analyzed genes and plasma levels of the studied proteins between VVs patients with varicosities localized in the great saphenous vein and in the small saphenous vein were investigated using the Mann Whitney U test and the Student’s *t*-test, depending on the normality of the data. No statistically significant associations were found; therefore, the expression of the analyzed genes and proteins is probably not significantly associated with the localization of varicosities.

## 3. Discussion

In the presented study, the dysregulations of the main factors that regulate angiogenesis and inflammation were investigated in patients with VVs. The expression levels of 18 genes (*ANGPT1*, *ANGPT2*, *CCL2*, *CCL5*, *CSF2*, *CXCL8*, *FGF2*, *IL1A*, *IL1B*, *IL6*, *PDGFA*, *PDGFB*, *TGFA*, *TGFB1*, *TNF*, *VEGFA*, *VEGFB*, and *VEGFC*) were analyzed in PBMC samples of VVs patients compared to controls. Furthermore, plasma levels of 6 proteins (ANGPT-1, ANGPT-2, TGF-alpha, TGF-beta 1, VEGF-A, and VEGF-C) were also compared between these groups of subjects.

A higher expression of *CCL5, PDGFA*, and *VEGFC*, as well as higher plasma levels of TGF-alpha, TGF-beta 1, and VEGF-A in the VV group, were found to be statistically significant. In turn, a statistically significant lower expression of *VEGFB* and plasma levels of VEGF-C were found in the VV group (Table 2 and Table 3, Figure 1 and Figure 2). These results suggest that changes in the activity of the signaling pathways associated with these factors could contribute to the pathogenesis of VVs.

Expression of inflammatory chemokine *CCL5* was shown in this study to be upregulated in the PBMC samples of the VV group vs. the control group. CCL5 belongs to the C-C motif chemokine family and is a key proinflammatory chemokine that has been shown to induce migration and recruitment of immune cells, such as T cells, dendritic cells, eosinophils, NK cells, mast cells, and basophils. The biological effects of CCL5 are mediated by its interaction with CCR1, CCR3, and CCR5 chemokine receptors on the cell surface [37]. CCL5 is also involved in the regulation of cell proliferation, apoptosis, invasion, division, metastasis, and inflammation by triggering various downstream pathways, including the PI3K/AKT, NF-kB, HIF-a, RAS-ERKMEK, JAK-STAT, and TGF-beta-Smad pathways [38]. At the early stages of atherosclerosis, CCL5 is secreted by activated platelets and promotes the adhesion of monocytes and neutrophils [39]. As this factor exerts chemoattractant properties, it could contribute to the leukocyte–endothelial interactions and leucocyte infiltration into the varicose venous wall, triggering inflammatory processes in response to abnormal venous flow. This is in agreement with previous studies on the role of inflammation in VVs development, which reported a systemic increase in leukocyte adhesion [40] and showed elevated expression of inflammatory markers in this disease [12].

Studies have shown that CCL5 is also involved in the regulation of angiogenesis; however, there is no consistency on whether its effect is pro- or antiangiogenic [37]. The plethora of biological processes and signaling pathways regulated by CCL5, as well as associated diseases, makes it difficult to predict the detailed role of the increased signaling of this factor in VVs. A general conclusion can be drawn that CCL5 appears to be implicated in a higher inflammatory state and dysregulated angiogenesis in VVs; however, further studies are needed to gain deeper insight into related downstream signaling pathways and exerted biological effects.

Interestingly, previous studies showed that the proangiogenic activity of *CCL5* is exhibited, at least partially, by potentiating VEGF signaling [41], raising this pathway as a potential contributor to VVs. This conclusion could be supported by other studies in which levels of VEGF, VEGF-A, and VEGFR2 were increased in the wall of varicose veins compared to the wall of normal ones [29,42,43,44]. In our study, similar results were obtained because higher levels of *VEGFC* in PBMC and VEGF-A in plasma were demonstrated in VVs patients; however, expression of *VEGFB* in PBMC and VEGF-C levels in plasma were lowered. These findings strongly suggest that altered VEGF signaling and regulated biological processes (including angiogenesis, infiltration of inflammatory cells, MMP activity, and maintenance of the integrity of the vascular wall) contribute to VVs. This conclusion could be further supported by studies on the effects of compression therapy in patients with CVD, where lowering of VEGF and other proinflammatory cytokines levels was observed [45,46].

Increased VEGF signaling in varicose veins can be transmitted through a VEGFR2 receptor that was previously shown to be upregulated in the VVs wall in comparison with the normal wall [29]. It could also suggest the role of glycocalyx constituents, which are involved in the mechanotransduction of shear stress, vascular inflammation, and angiogenesis by influencing VEGF-A and VEGFR2 signaling. Glycocalyx components play a crucial role in the pathology of diseases related to high blood pressure and disturbed blood flow, such as VVs. However, many components of the glycocalyx have a distinct function in promoting or inhibiting angiogenesis, depending on whether they are present in the endothelial cell or shed [10,47,48,49]. Therefore, the increased expression of VEGF-A and VEGFR2 previously demonstrated in patients with VVs suggests the proangiogenic composition of the glycocalyx structure, potentially determined by the shedding of glycocalyx caused by venous hypertension and inflammation, the primary hallmarks of VVs and more advanced symptoms of chronic venous disease. However, to gain more detailed insight into the implications of glycocalyx composition in the pathophysiology of VVs, more studies focused on this aspect are required.

The changes in VEGFC expression found in our study in VVs patients may suggest altered regulation of lymphangiogenesis in these patients since VEGFC-mediated VEGFR3 signaling is crucial for this process [50,51,52]. This conclusion can be supported by the other study, in which a higher expression of other genes that regulate this process was previously associated with chronic venous disease in pregnant women [53]. Although high expression of VEGFR3 is a marker of lymphatic vessels and especially lymphatic valves, this receptor is also expressed in developing venous valves [54]. Therefore, dysregulations in VEGF-C presented in our study may also indicate the role of regulation of the development of venous valves in VVs; however, the impact on this process is difficult to estimate because in PBMC, the expression of VEGFC was found to increase, but in plasma, it was reduced. Further studies are needed to elucidate the regulation of lymphangiogenesis in patients with this disease.

In our study, higher levels of *CCL5* and *PDGFA* mRNA were found in the VV group. Increased levels of proteins encoded by these genes were shown in previous studies in blood from the varicose vein site compared to systemic concentrations [55,56]. A higher expression of *PDGFA* in patients with VVs compared to control subjects could be another hallmark of increased angiogenesis status in VVs since *PDGFA* is known as a strong stimulator of this process [57,58].

Regarding the TGF family factors, both TGF-alpha and TGF-beta 1 encoding genes had similar expression in PBMC in the study groups, but in plasma, the concentrations were higher in the VV group. In the research literature, studies focused on the role of TGF-alpha in VVs are limited, whereas more studies investigated the role of TGF-beta 1 in VVs. Similarly to PBMC in our study, previous studies described no significant change in TGF-beta 1 expression in varicose vein samples compared to normal veins. However, the protein content of the active form of this factor was found to be decreased in varicose vein samples, while the total content of TGF-beta 1 was shown to increase in VVs, but the difference was not always statistically significant [59,60]. On the contrary, Pascual and colleagues demonstrated a significant elevation of the active form of TGF-beta 1 in VVs, which was positively correlated with the age of the studied subjects [61]. A higher expression of this factor in VVs samples was also correlated with the presence of macrophages and overproduction of iNOS [62,63]. The presented discrepancies in the results obtained in the previous studies make the discussion about the association between elevated plasma levels of TGF-beta 1 in VVs patients and the effect on TGF-beta 1 signaling in VVs pathology difficult. The results of the conducted studies have suggested that the processes that contribute to vascular remodeling observed during the development of varicose veins could be an effect of altered TGF-beta 1 signaling [64]; however, further studies are required to explore this association and to uncover whether the altered TGF-beta 1 signaling in VVs is reflected by circulatory markers.

The presented study focused on circulatory markers of VVs, and the study design raises the question of to what extent the changed levels of such markers reflect the pathological mechanisms ongoing in the vascular wall during VVs development. Interactions between circulatory cells and pathologically changed vascular walls could cause the acquisition of new features, which could be used to detect the disease. Such features could include elevated expression of *CCL5* because its dysregulations were observed not only in PBMC (our study) but also in varicose vein samples [65]. Furthermore, some proteins and other factors could be secreted from the pathologically changed vascular site and could be reliable indicators of the disease. Such indicators could be higher plasma levels of VEGF-A, which could illustrate previously described higher VEGF-A activity in VVs.

Several limitations were identified for the presented study. Performed investigations regarded only selected main regulators of inflammation and angiogenesis, but these processes are very complex and involve many receptors and downstream effectors, which analysis is out of the scope of this study. The presented study focused only on positive regulators of inflammation and angiogenesis since there is evidence that these processes are enhanced in VVs, but the analysis of inhibitors of these processes, which could also contribute to the disease, was not included. Furthermore, it is not clear whether the presented dysregulations are causative or a secondary hallmark of a disease. Due to limited resources, plasma protein levels were not measured for all factors used for gene expression analysis. The number of the analyzed samples was limited, not fully balanced between groups, and was not sufficient to draw a definitive conclusion about the observed associations, raising the need for validation of the obtained results in studies with larger cohorts. Finally, the conclusions and hypotheses stated in the discussion section have a speculative character and need to be further investigated in future studies. However, despite these limitations, the results obtained provide a primary image of the changes in the expression of the main regulators of inflammation and angiogenesis associated with VV. The dysregulations that were presented, after confirmation in further in-depth studies, could potentially be used as a diagnostic or therapeutic target for better clinical care of patients with VVs.

## 4. Materials and Methods

### 4.1. Study Participants

The study was carried out according to the Declaration of Helsinki, and the study procedure was approved by the Bioethics Committee at the Medical University of Lublin (decision No. KE-10-0254/148/2021). The study population included two groups of subjects: 40 patients with VVs (VV group) and 24 healthy volunteers (control group). Qualification of the study participants was obtained from the Chair and Department of Vascular Surgery and Angiology of the Medical University of Lublin by a vascular surgeon (M.F.). Informed and signed consent was obtained from all study subjects.

The VV group consisted of non-smoking patients diagnosed with VVs on the basis of the medical interview, visual inspection of lower extremities, physical examinations using a tourniquet test and auscultation, as well as vascular imaging by color duplex ultrasonography. The application of various examination methods ensured a precise diagnosis and accurate anatomical localization of abnormalities. The included patients were classified according to the CEAP classification [18] into categories: C2 (clinical presentation as varicose veins), As (symptoms of superficial veins), Ep (primary etiology), and Pr (reflux pathophysiology).

The exclusion criteria were as follows: previous vascular surgery of the lower limbs, deep vein insufficiency, deep vein thrombosis, lower extremity artery disease, coronary artery disease, cerebrovascular disease, aneurismal disease, myocardial infarction, hypertension, stroke, type 2 diabetes mellitus, and pregnancy.

The control group included 24 healthy, non-smoking volunteers with neither hemodynamic blood flow disturbances nor abnormalities in vascular morphology observed in the lower limb region during the visual examination and color flow duplex ultrasound scanning.

All participants did not use acetylsalicylic acid, clopidogrel, fibrates, Ca^2+^ channel blockers, and angiotensin-converting enzyme inhibitors in their medication history. The demographic and clinical characteristics of the studied subjects are presented in Table 1.

### 4.2. Real-Time PCR Experiments

The real-time PCR method was used to determine expression levels of 18 genes associated with the regulation of angiogenesis and inflammation (*ANGPT1*, *ANGPT2*, *CCL2*, *CCL5*, *CSF2*, *CXCL8*, *FGF2*, *IL1A*, *IL1B*, *IL6*, *PDGFA*, *PDGFB*, *TGFA*, *TGFB1*, *TNF*, *VEGFA*, *VEGFB*, and *VEGFC*) in the studied subjects. Peripheral blood mononuclear cells (PBMC) were used as biological material and were isolated from the venous blood samples of the included subjects using gradient centrifugation with Gradisol L reagent (Aqua-Med, Łódź, Poland) according to the standard procedure [66].

Subsequently, the PBMC samples were subjected to total RNA extraction using TRI Reagent Solution (Ambion, Austin, TX, USA), according to the manufacturer’s procedure. The quality and quality of the total RNA samples were assessed using the NanoDrop ND-1000 spectrophotometer (Thermo Fisher Scientific, Waltham, MA, USA) and the Agilent 2100 Bioanalyzer with the Agilent RNA 6000 Pico Kit (Agilent Technologies, Santa Clara, CA, USA). For further experiments, the total RNA samples with a 260/280 ratio higher than 1.8 and RNA integrity number greater than 7 were used.

The total RNA samples were then reverse transcribed using the High Capacity cDNA Reverse Transcription Kit (Applied Biosystems, Foster City, CA, USA) according to the manufacturer’s protocol. Before the reaction, the RNA samples were diluted to 100 ng/µL. The real-time PCR reactions that contain 2 µL of cDNA, 10 µL of TaqMan Gene Expression Master Mix (Applied Biosystems, Foster City, CA, USA), 7 µL of nuclease-free water, and 1 µL of TaqMan Gene Expression Assay specific to the target gene (Applied Biosystems, Foster City, CA, USA) were prepared in 96-well plates. Information about used assays is provided in Table 5. *GAPDH* was used as an endogenous control, as well as blank reactions (without cDNA) were performed to detect potential contamination by foreign DNA. The real-time PCR reactions were carried out in triplicates.

The amplification of target genes was carried out using the 7900HT Real-Time Fast System (Applied Biosystems, Foster City, CA, USA) and the following steps: initial denaturation (95 °C for 10 min) and 40 amplification cycles (95 °C for 15 s and 60 °C for 1 min in each cycle).

The expression levels of the analyzed genes were determined as Ct values (the numbers of amplification cycles achieved at the intersection between an amplification curve and a threshold line defining the linear phase of the signal growth) using the ExpressionSuite v1.3 software (Life Technologies Corporation, Carlsbad, CA, USA).

All subsequent steps of the analysis were performed using R 4.3.1 programming software (https://www.r-project.org, accessed on 14 December 2023). The Ct values with low reliability (higher than 35, flagged as ‘undetermined’ or ‘inconclusive’, or with the parameter AMPSCORE < 1) were filtered out. The amounts of data filtered and retained for the analysis of each gene are presented in Appendix A. For four genes (*ANGPT2*, *CSF2*, *IL1A*, and *IL6*), Ct data were filtered in more than half of the subjects in at least one of the study groups; therefore, these genes were excluded from the study. A similar criterion was used for samples: those with filtered data for more than half of the analyzed genes were considered unreliable. Using this criterion, two samples in the VV group were removed from the data (Appendix A).

After data filtering, the dataset contained 6.7% missing values; therefore, a data imputation procedure using a multivariate linear regression method was implemented. In this procedure, the missing values were filled using genes with complete data.

The complete Ct dataset was further analyzed using the delta Ct method for relative quantification [35,36]. For each sample, the Ct values of target genes were normalized using the Ct values of the endogenous control (*GAPDH*) by subtracting the mean Ct value for the *GAPDH* replicates from the mean Ct value for the target gene replicates, obtaining delta Ct (dCt) values. Subsequently, the dCt values were transformed using a 2^−dCt^ formula to achieve a linear form of data. The uniformity of the transformed data was evaluated using hierarchical clustering (Appendix A) and PCA analysis (Appendix A). Performed assessments showed that eight samples had an outlier character and thus were excluded from the data. The retained data has good homogeneity and was used for further steps of the analysis.

Differences in the expression of target genes between the VV and control groups were calculated by dividing the mean of the transformed dCt values of the VV group by the mean of the transformed dCt values of the control group, obtaining fold change values. Statistical analysis of the differences in gene expression between VV and control groups was performed according to the procedure described in Section 4.4.

### 4.3. ELISA Experiments

The concentrations of ANGPT-1, ANGPT-2, TGF-alpha, TGF-beta 1, VEGF-A, and VEGF-C proteins in plasma samples collected from 40 patients with AAA (AAA group) and 20 healthy volunteers (control group) were determined using the enzyme-linked immunosorbent assay (ELISA) method. Plasma was separated from venous blood samples by centrifugation (2000× *g* for 10 min), aliquoted, and stored at −80 °C. Measurements were performed using commercially available ELISA kits purchased from Biorbyt (Cambridge, UK) and according to the manufacturer’s instructions (Table 6).

Prior to the analysis, the thawed aliquots of plasma samples were centrifuged (2000× *g* for 10 min at 4 °C) to remove residual platelets and cells. To ensure that the protein concentration obtained in the analysis will fall near the middle of the range in the standard curve, the concentrations of target proteins were primarily estimated on the basis of the literature data, and, if necessary, the samples were diluted using a diluent buffer provided with the ELISA kit. For each ELISA experiment, eight standard concentrations were performed to draw the standard curve, as well as blank wells that contained a dilution buffer. When indicated in the experimental procedure, the ELISA plates were incubated using the DTS-2 incubator (ELMI, Riga, Latvia) and read at the appropriate wavelength using the Synergy H1 microplate reader (BioTek, Winooski, VT, USA).

Analysis of raw data was performed using Gen5 version 3.10 software (BioTek, Winooski, VT, USA). The background absorbance values obtained from the blank wells were subtracted from the values obtained from other wells. The standard curve was generated to present the arrangement of the relative absorbance value of each standard and the corresponding concentration of the standard solution. The protein concentrations in the samples were interpolated from the standard curve. If the samples were diluted, the concentrations obtained from interpolation were multiplied by the dilution factor to obtain the final concentration.

Quality control of the obtained data included the analysis of the number of samples with detected (within the quantification range) and not detected (below the quantification range) concentrations of analyzed proteins (Appendix A). The consistency of the data was assessed by hierarchical clustering (Appendix A) and in the PCA plot (Appendix A), and two outlier samples were identified. Differences in plasma protein levels between the VV and control groups were analyzed using the appropriate statistical tests (see the next paragraph).

### 4.4. Statistical Analysis

The statistical analysis was performed using the R 4.3.1 environment (https://www.r-project.org/, accessed on 14 December 2023), and appropriate statistical methods were selected depending on the type and distribution of the analyzed variables.

Correlation analysis, as well as univariate and multivariate linear regression, was used to investigate relationships between continuous variables. Correlation analysis was performed using the Spearman rank correlation test implemented in the Hmisc 5.1-1 package (https://cran.r-project.org/web/packages/Hmisc/index.html, accessed on 22 January 2024), while linear regression models were constructed using the lm base function in R.

Univariate and multivariate logistic regression, as well as two two-sided statistical tests (Student’s *t*-test and Mann Whitney U test), were used to analyze associations between continuous and categorical variables. The selection of the proper test depended on the normality of the data, assessed using the Shapiro-Wilk test (shapiro.test function in R). If the distributions of the analyzed variables in both compared groups were defined as normal (*p* > 0.05 in the Shapiro-Wilk test), the parametric Student’s *t*-test was used, while if the distributions of the dCt values in at least one of the compared groups were defined as not normal (*p* < 0.05 in the Shapiro-Wilk test), the non-parametric Mann-Whitney U test was used. The two-sided Student’s *t*-test was performed using the *t*-test function in R, while the two-sided Mann Whitney U test was carried out using the wilcox.test function in R. In turn, logistic regression models were constructed using the glm base function in R.

The Fisher’s exact test (the fisher.test function in R) was used to assess the relationships between categorical variables.

The pROC package 1.18.5 [67] (https://cran.r-project.org/web/packages/pROC/index.html, accessed on 22 January 2024) was used to perform the ROC analysis and draw ROC plots.

The results obtained with *p* < 0.05 were considered statistically significant. Visualizations were generated using the ggplot2 3.4.4 package (https://ggplot2.tidyverse.org/, accessed on 22 January 2024) unless otherwise noted.

## 5. Conclusions

Demonstrated dysregulations in key regulators of angiogenesis and inflammation in patients with VVs indicate that the abnormal course of these processes could contribute to this disease. Higher expression of *CCL5*, *PDGFA*, and *VEGFC* in PBMC, as well as higher plasma levels of TGF-alpha, TGF-beta 1, and VEGF-A, suggest enhanced angiogenesis and inflammation pathways involved in the development of VVs. The presented dysregulations can be a promising diagnostic or therapeutic target; however, further studies are needed to validate the obtained results.

## Figures and Tables

**Figure 1 ijms-25-06785-f001:**
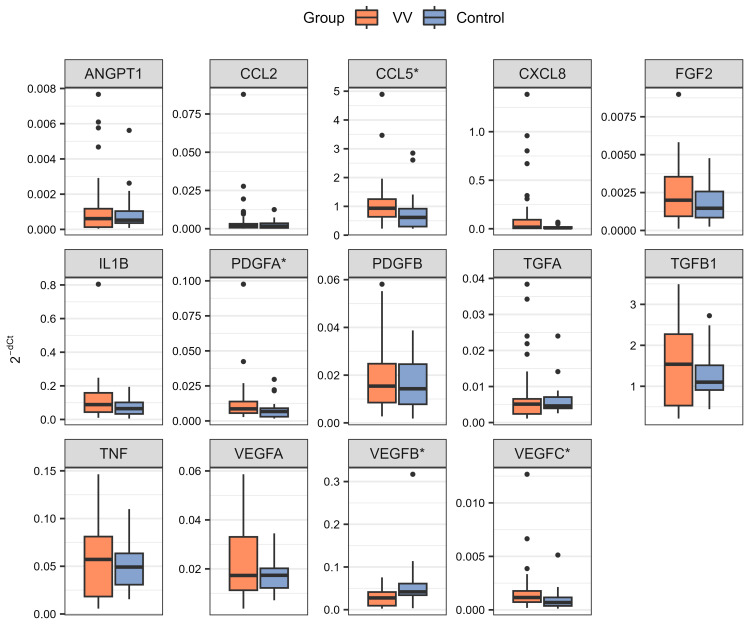
The distribution of 2^−dCt^ values was calculated for 14 analyzed genes in the VV and control groups. Whiskers reach the most distant point in the doubled interquartile range (samples located outside the whiskers are marked as a round point), boxes range between 25% and 75% quartile, and horizontal lines inside boxes mark the median value. *—genes with statistically significant differences between compared groups (*p* < 0.05, see Table 2 for exact *p* values), VV—the group of patients with varicose veins, Control—the group of control subjects.

**Figure 2 ijms-25-06785-f002:**
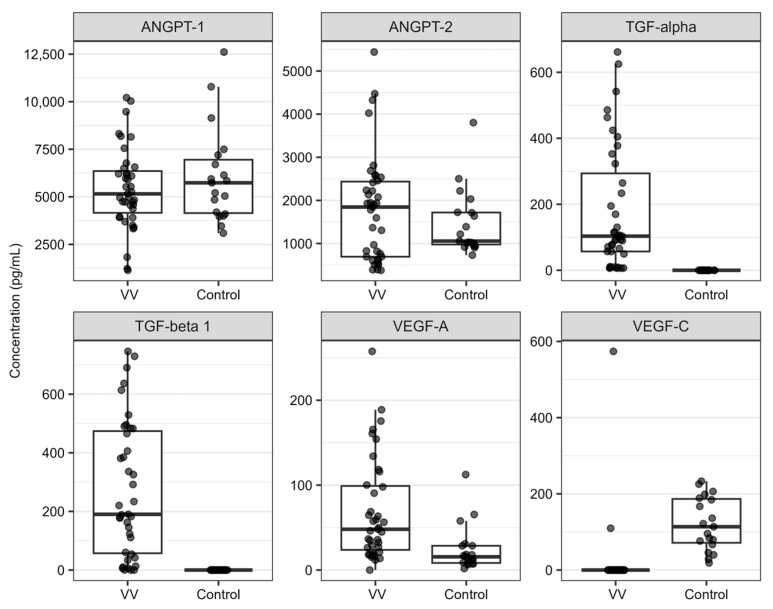
Distributions of plasma levels obtained for analyzed proteins in patients with VVs (VV group) and healthy controls (control group). Whiskers reach the most distant point in the 1.5 interquartile range, boxes range between 25% and 75% quartile, and horizontal lines inside boxes mark the median value.

**Figure 3 ijms-25-06785-f003:**
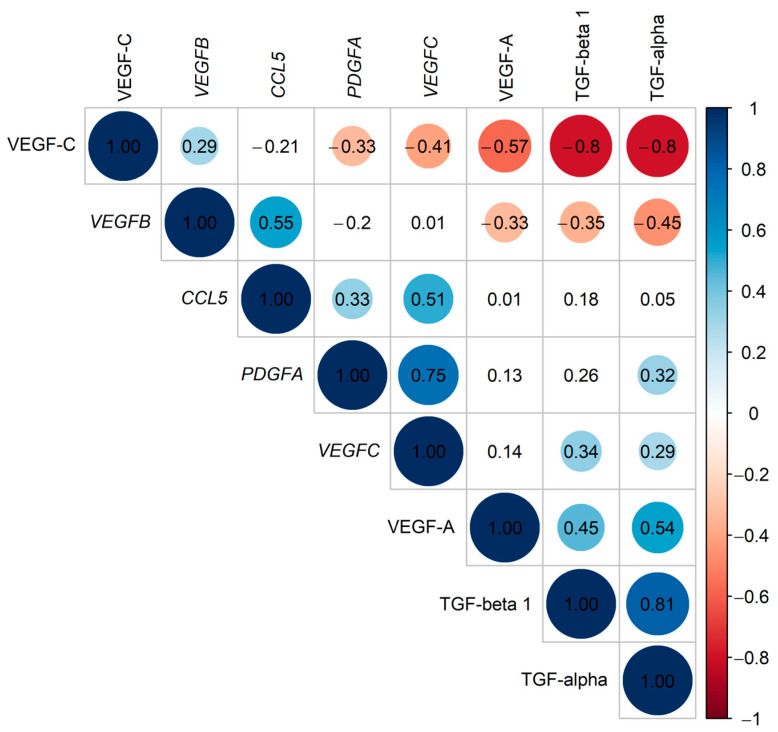
Correlation coefficients obtained for expression levels of 4 selected genes (*CCL5, PDGFA, VEGFB*, and *VEGFC*) and plasma levels of 4 selected proteins (TGF-alpha, TGF-beta 1, VEGF-A, and VEGF-C), calculated using Spearman rank correlation test. The plot was generated using the corrplot 0.92 package in R. The colored circles mark correlation coefficients with statistical significance (*p* < 0.05).

**Table 1 ijms-25-06785-t001:** Demographical and clinical data of study subjects.

Characteristic	VV Group (n = 40)	Control Group (n = 24)	*p* ^1^
Age	53.7 ± 8.00 (39–72)	55.6 ± 9.13 (35–73)	>0.05
Sex male/female	27 (67.5%)/13 (32.5%)	14 (58.3%)/10 (41.7%)	>0.05
Body mass index (BMI)	25.8 ± 3.40 (17.6–32.5)	25.2 ± 3.09 (21.2–32.9)	>0.05
Hypertension	1 (2.5%)	4 (16.7%)	>0.05
Hypercholesterolemia	9 (22.5%)	7 (29.2%)	>0.05
LDL (mg/dL)	96.6 ± 14.2 (71–121)	102 ± 9.9 (84–117)	>0.05
HDL (mg/dL)	41.7 ± 3.52 (33–48)	41.2 ± 2.88 (35–47)	>0.05
Cholesterol (mg/dL)	203 ± 18.2 (167–242)	191 ± 9.14 (178–204)	4.501 × 10^−3^
Creatinine (mg/dL)	0.63 ± 0.13 (0.34–0.89)	0.8 ± 0.13 (0.45–1.03)	7.378 × 10^−6^
Urea (mg/dL)	31.2 ± 6.5 (21–45)	36.3 ± 2.57 (31–41)	4.584 × 10^−5^
C-reactive protein (mg/L)	2.5 ± 1.1 (0.8–5.2)	2.5 ± 0.9 (1.1–4.3)	>0.05
Fibrinogen (mg/dL)	163 ± 33.5 (109–261)	195 ± 40.7 (112–278)	5.742 × 10^−4^
Homocysteine (µmol/L)	6.4 ± 1.4 (3.9–8.9)	7.2 ± 1.3 (5.1–10.8)	2.514 × 10^−2^
Medication with statins	8 (20%)	7 (29.2)	>0.05
Medication with beta-adrenergic blockers	1 (2.5%)	4 (16.7)	>0.05
Medication with diosmin	24 (60%)	0 (0%)	3.663 × 10^−7^

^1^ Statistical significance of differences between the group of patients with varicose veins (VV group) and the group of control subjects (control group) was calculated for continuous-type variables (age, BMI, and biochemical blood parameters) using the two-sided Student’s *t*-test or the two-sided Mann Whitney U test (depending on the normality of the data), while for categorical-type variables (sex, hypertension, and hypercholesterolemia), the two-sided Fisher’s exact test was used. Continuous-type variables are presented as mean ± SD and range in brackets. Categorical-type variables are presented as counts and percentages in brackets.

**Table 2 ijms-25-06785-t002:** Results of differential gene expression analysis between VV and control groups.

Gene Symbol	Gene Name	Differential Expression		ROC	Univariate Logistic Regression	
		Fold Change	*p*	ROC-AUC	OR	*p*
*ANGPT1*	Angiopoietin 1	1.430	0.847	0.484	1.209	0.384
*CCL2*	C-C motif chemokine ligand 2	2.411	0.755	0.526	1.311	0.345
*CCL5*	C-C motif chemokine ligand 5	1.407	0.045	0.661	1.710	0.193
*CXCL8*	C-X-C motif chemokine ligand 8	10.067	0.197	0.604	2.814	0.188
*FGF2*	Fibroblast growth factor 2	1.405	0.243	0.595	1.838	0.139
*IL1B*	Interleukin 1 beta	1.642	0.172	0.610	1.999	0.144
*PDGFA*	Platelet derived growth factor subunit A	1.775	0.029	0.676	1.981	0.136
*PDGFB*	Platelet derived growth factor subunit B	1.180	0.768	0.525	1.368	0.424
*TGFA*	Transforming growth factor alpha	1.280	0.445	0.438	1.252	0.425
*TGFB1*	Transforming growth factor beta 1	1.175	0.849	0.516	1.368	0.359
*TNF*	Tumor necrosis factor	1.145	0.795	0.522	1.390	0.453
*VEGFA*	Vascular endothelial growth factor A	1.260	0.591	0.456	2.042	0.172
*VEGFB*	Vascular endothelial growth factor B	0.481	0.002	0.748	0.243	0.011
*VEGFC*	Vascular endothelial growth factor C	1.905	0.020	0.686	1.910	0.139

Gene symbols and gene names are provided in accordance with actual nomenclature in the HUGO Gene Nomenclature Committee (HGNC) (https://www.genenames.org/, accessed on 26 January 2024). OR—odds ratio, ROC-AUC—area under receiver operating characteristics curve.

**Table 3 ijms-25-06785-t003:** Differences in analyzed plasma protein levels between the VV and control group.

Protein Symbol	Protein Name	Mean Concentration (pg/mL)		*p*	AUC-ROC
		VV	Control		
ANGPT-1	Angiopoietin-1	5439.50 ± 2123.74	6078.81 ± 2511.29	0.532	0.552
ANGPT-2	Angiopoietin-2	1792.27 ± 1241.33	1463.36 ± 757.34	0.669	0.536
TGF-alpha	Protransforming growth factor alpha	181.52 ± 187.30	0.00 ± 0.00	4.37 × 10^−10^	1.000
TGF-beta 1	Transforming growth factor beta-1 proprotein	272.53 ± 230.52	0.00 ± 0.00	1.07 × 10^−9^	0.987
VEGF-A	Vascular endothelial growth factor A	69.00 ± 60.46	24.59 ± 27.13	1.33 × 10^−4^	0.800
VEGF-C	Vascular endothelial growth factor C	17.53 ± 93.10	121.58 ± 69.84	4.57 × 10^−11^	0.962

The protein names provided are in accordance with the UniProt database (release 2024_01, https://www.uniprot.org/, accessed on 26 January 2024). *p*—statistical significance was calculated using a two-sided Mann-Whitney U test; ROC-AUC was the area under the receiver operating characteristics curve.

**Table 4 ijms-25-06785-t004:** Statistically significant correlations (*p* < 0.05) between selected genes or proteins and blood biochemical parameters.

Correlated Variables	Correlation		Univariate Linear Regression		
	R	*p*	β	*p*	R^2^
*CCL5*—urea	−0.42	1.66 × 10^−3^	−4.98 × 10^−2^	1.87 × 10^−2^	0.10
*PDGFA*—urea	−0.32	1.73 × 10^−2^	−7.56 × 10^−4^	3.46 × 10^−2^	0.08
*VEGFC*—urea	−0.40	2.75 × 10^−3^	−1.18 × 10^−4^	1.54 × 10^−2^	0.11
VEGF-C—creatinine	0.55	9.10 × 10^−6^	186.2	2.71 × 10^−2^	0.08
TGF-alpha—creatinine	−0.49	1.10 × 10^−4^	−328.2	2.84 × 10^−2^	0.08
TGF-beta 1—fibrinogen	−0.47	1.90 × 10^−4^	−2.173	4.95 × 10^−3^	0.13
VEGF-A—creatinine	−0.45	3.58 × 10^−4^	−189.7	2.47 × 10^−5^	0.27
VEGF-C—cholesterol	−0.35	7.50 × 10^−3^	−1.591	3.86 × 10^−2^	0.07
VEGF-A—homocysteine	−0.29	2.82 × 10^−2^	−13.20	1.91 × 10^−2^	0.09

R—correlation coefficient, β—regression coefficient, R^2^—determination coefficient.

**Table 5 ijms-25-06785-t005:** List of the TaqMan Gene Expression Assays used for the study.

Assay ID	Gene Symbol	Gene Name	Amplicon Length
Hs00919201_m1	*ANGPT1*	Angiopoietin 1	119
Hs00169867_m1	*ANGPT2*	Angiopoietin 2	73
Hs00234140_m1	*CCL2*	C-C motif chemokine ligand 2	101
Hs99999048_m1	*CCL5*	C-C motif chemokine ligand 5	98
Hs00929873_m1	*CSF2*	Colony stimulating factor 2	85
Hs00174103_m1	*CXCL8*	C-X-C motif chemokine ligand 8	101
Hs99999905_m1	*GAPDH*	Glyceraldehyde-3-phosphate dehydrogenase	122
Hs00174092_m1	*IL1A*	Interleukin 1 alpha	69
Hs01555410_m1	*IL1B*	Interleukin 1 beta	91
Hs00174131_m1	*IL6*	Interleukin 6	95
Hs00266645_m1	*FGF2*	Fibroblast growth factor 2	82
Hs00234994_m1	*PDGFA*	Platelet derived growth factor subunit A	93
Hs00966522_m1	*PDGFB*	Platelet derived growth factor subunit B	56
Hs00608187_m1	*TGFA*	Transforming growth factor alpha	70
Hs00998133_m1	*TGFB1*	Transforming growth factor beta 1	57
Hs00174128_m1	*TNF*	Tumor necrosis factor	80
Hs00900055_m1	*VEGFA*	Vascular endothelial growth factor A	59
Hs00173634_m1	*VEGFB*	Vascular endothelial growth factor B	69
Hs01099203_m1	*VEGFC*	Vascular endothelial growth factor C	66

Gene symbols and gene names are provided in accordance with actual nomenclature in the HUGO Gene Nomenclature Committee (HGNC) (https://www.genenames.org/, accessed on 26 January 2024).

**Table 6 ijms-25-06785-t006:** ELISA kits were used for the study.

ELISA kit ID	Protein Symbol	Protein Name	Quantitative Range (pg/mL)
orb138056	ANGPT-1	Angiopoietin-1	156.25–10,000
orb146693	ANGPT-2	Angiopoietin-2	156.25–10,000
orb50169	TGF-alpha	Protransforming growth factor alpha	15.625–1000
orb50103	TGF-beta 1	Transforming growth factor beta 1 proprotein	15.625–1000
orb50119	VEGF-A	Vascular endothelial growth factor A	31.25–2000
orb50131	VEGF-C	Vascular endothelial growth factor C	62.5–4000

The protein names provided are in accordance with the actual nomenclature in the Uniprot database (https://www.uniprot.org/, accessed on 26 January 2024).

## Data Availability

The data used for this study are openly available in the FigShare repository: https://doi.org/10.6084/m9.figshare.26065462.v1.

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
