# Peer review of "Key Regulators of Angiogenesis and Inflammation Are Dysregulated in Patients with Varicose Veins"

_ijms, 2024, doi:10.3390/ijms25126785_

Round 1
Reviewer 1 Report
Comments and Suggestions for Authors
Thank you for this manuscript submission, novel in approach.
Can you clarify the potential for;
1) is there correlation with endothelial glycocalyx regulation and alterations/shedding? Endothelial glycocalyx now well recognized as a key component of angiogenesis and shedding/dysfunction/inflammation in venous hypertension
2) is there any correlation with lymphangiogenesis/lymphatic dysfunction; VEGF-C noted in your identification; venous disease associated with C3 and beyond for venous insufficiency -
Reviewer 2 Report
Comments and Suggestions for Authors
Content suggestions:
1. There is relatively high percent match 45%. Further negative is the use of lower number of controls than the patients included in the study.
2. For the completeness, I would like to kindly ask the Authors to add information about family history, drug history and gynecological history of the patients.
I sincerely appreciate the effort of the Authors to perform such an important study, as varicose veins are the significant risk factor of life-threatening venous thromboembolism. The manuscript is written on a high scientific level with the use of sophisticated statistical methods and illustration of the results in several figures and tables.
